# Advances in Portable Atom Interferometry-Based Gravity Sensing

**DOI:** 10.3390/s23177651

**Published:** 2023-09-04

**Authors:** Jamie Vovrosh, Andrei Dragomir, Ben Stray, Daniel Boddice

**Affiliations:** 1School of Physics and Astronomy, University of Birmingham, Birmingham B15 2TT, UK; javovrosh@qinetiq.com (J.V.);; 2QinetiQ, Malvern Technology Centre, St. Andrews Road, Malvern, Worcestershire WR14 3PS, UK; 3Aquark Technologies, Abbey Park Industrial Estate, Romsey SO51 9AQ, UK; 4School of Engineering, University of Birmingham, Birmingham B15 2TT, UK

**Keywords:** atom interferometry, gravity sensing, quantum sensing, gravity, quantum sensors, quantum technology, cold atoms, 03.75.Dg, 06.20.−f, 07.07.Df, 91.10.Op, 93.85.+q

## Abstract

Gravity sensing is a valuable technique used for several applications, including fundamental physics, civil engineering, metrology, geology, and resource exploration. While classical gravimeters have proven useful, they face limitations, such as mechanical wear on the test masses, resulting in drift, and limited measurement speeds, hindering their use for long-term monitoring, as well as the need to average out microseismic vibrations, limiting their speed of data acquisition. Emerging sensors based on atom interferometry for gravity measurements could offer promising solutions to these limitations, and are currently advancing towards portable devices for real-world applications. This article provides a brief state-of-the-art review of portable atom interferometry-based quantum sensors and provides a perspective on routes towards improved sensors.

## 1. Introduction

Gravity sensors are employed across various fields, including metrology [1], civil engineering [2,3], geology [4], archaeology [5], environmental monitoring [6,7,8], carbon capture and storage [9], and resource exploration [10]. Although traditional gravimeters are valuable sensors [11], they are limited by wear on their test masses, instrumental drift, microseismic vibrations, and variations in absolute values between different sensors, hindering their use for long-term observations and in arrays of multiple sensors [11,12,13]. A new generation of gravity sensors based on cold atoms, referred to as cold atom gravimeters (CAGs), could offer a solution to these limitations [14], providing consistency of measurements both spatially and temporally.

CAGs utilise cold atoms as test masses and measure gravity through atom interferometry. The first atom interferometer was at Stanford University in 1991, Kasevich et al. [15] and used to perform gravity measurements. Since this initial demonstration there has been an international wave of atom interferometry and related research [14,16], which has resulted in atom interferometers having become extremely sensitive devices achieving measurement sensitivities of up to 2.2 μGal/$\sqrt{Hz}$ (1 Gal = 1 cm/s${}^{2}$) and resolutions of 0.08 μGal after integration times of 2000 s [17].

CAGs offer several advantages over classical gravimeters. They contain no macroscopic moving components, allowing for long-term continuous measurements and high sampling rates without degradation from thermal or wear effects [18]. The precision and stability of CAGs are also enhanced by the use of atom standards for reference enabling low drift. These benefits have sparked interest in using portable CAGs for various applications, such as civil engineering [19], atmospheric drag measurements [20], carbon capture and storage monitoring [21], climate change monitoring [22,23], navigation [18,23,24,25,26,27,28], geohazard monitoring [29], and military applications [30,31,32].

This article provides a brief review of the current advancements in developing portable atom interferometry-based gravity sensors designed for use in several applications, as well a providing a brief overview of different technological and physical techniques that could be used to produce improved CAGs and a more mature CAG market.

## 2. Operating Principle

An atom interferometer in its simplest form can be implemented by dropping a cloud of cold atoms in a vacuum [15]. These atoms are initially trapped, cooled, and prepared in a single atomic state before being allowed to fall under gravity, during which they are subjected to a sequence of three laser pulses, as shown in Figure 1.

The first laser pulse in this sequence puts the atoms into an equal superposition of two states; after a time, T, another laser pulse is applied which swaps the states, owing to atoms absorbing and emitting a photon, respectively. The associated momentum transfers make the atomic trajectories converge, such that they intersect at the same point in space after a further time T. A final laser pulse then closes the interferometer sequence, allowing for interference between the two different trajectories taken by the matter waves associated with the atoms. The gravitational acceleration, g, experienced by the atoms results in a phase shift Δϕ=2k${}_{eff}$gT${}^{2}$, where k${}_{eff}$ is the effective wavevector of the light, which determines the momentum transfer. Δϕ can be read out from the atom interferometer by counting the number of excited-state atoms versus the number of ground-state atoms at the end of the sequence. The shot noise limited sensitivity for an atom interferometer is given by
(1)$$\frac{\Delta g}{g}=\frac{1}{K_{eff}T^{2}\sqrt{N}}$$
where N is the number of atoms that undergo the atom interferometry process.

It is possible to create a cold atom gravity gradiometer (CAGG), by utilising two atom interferometers spaced a distance apart, with the interferometry beams measuring on the same axis.

More detailed information about the underlying theory and operating principles can be found in references [18,33,34,35], while more information on the underpinning technologies can be found in references [36,37,38].

## 3. Progress to Date

Lab-based systems have been used in a variety of research areas including metrology and fundamental physics, where they have been used for measurements of the fine structure constant [39,40,41], the Newtonian gravitational constant [42,43], testing of general relativity [44,45], the isotropy of gravitational interaction [46,47], the equivalence principle [48,49,50,51] and in the search for new forces [52,53,54].

The success of the lab-based system has prompted the desire to transition these devices into portable sensors that can be used in a variety of applications [14,55,56]. To achieve this atom interferometers are evolving from complicated setups in research laboratories to practical, and in some cases transportable, instruments. As part of this process, the size, weight, and power (SWaP) of these sensors have had to be reduced along with improving robustness [57] to environmental and motional effects and usability, while also maintaining sufficient sensitivity to be useful in the desired application. Examples of signal sizes from different applications can be seen in Table 1.

Several portable CAGs have been developed to date, with earlier prototypes performing demonstrations outside [81,82,83] in lifts [84] or underground laboratories [85]. More recently, however, portable CAGs have been used in several real-world applications. A brief summary of the different demonstrations to date will follow.

### 3.1. Metrology

Within metrology, lab-based and portable CAGs have been used to better define the fundamental constants, with portable systems being used for applications where the instrument needs to be moved to work in tandem with another instrument which may be contained in a different laboratory far from the location of static CAGs. For example, in 2017, a portable CAG was used to determine the Planck constant using the LNE Kibble balance in air [86].

CAGs have participated in several comparisons of absolute gravimeters, comparing favourably with other technologies. The cold atom gravimeter (LNE-SYRTE) has been participating in international comparisons of absolute gravimeters since 2009 [87,88,89]. Another CAG participated in the first Asia-Pacific Comparison of Absolute Gravimeters, hosted by the National Institute of Metrology of China from December 2015 to March 2016 [90]. In 2017, six CAGs took part in the 10th International Comparison of Absolute Gravimeters, from institutions including the Zhejiang University of Technology, the Huazhong University of Science and Technology, the University of Science and Technology of China, the Changcheng Institute of Metrology and Measurement Beijing, the National Institute of Metrology and the Wuhan Institute of Physics and Mathematics. Several results from these sensors were accepted by the committee and showed performance comparable to classical corner cube gravimeters [91,92], showing the current generation of portable CAGs is suitable for use in metrological applications. Without the limitations of mechanical wear and with the potential for the sensitivity to improve further in the future, it is expected the use of CAGs will increase in metrological applications as they become more readily available commercially.

### 3.2. Environmental Monitoring

Portable CAGs have been demonstrated to be able to monitor tidal gravity variations [93,94]. For example, the CAG from the University of California [93] in 2017 was used to measure tidal gravity measurements and was capable of detecting both the effects of local and global tidal gravity variation.

CAGs have been used for seismic monitoring, for example, on the 28th of September 2013 a CAG from Zhejiang University detected a seismic wave, which originated in Pakistan from a 7.2 magnitude earthquake [95]. Other examples include the CAG from the University of California which was used to detect earthquakes in Berkeley originating from other parts of the world [93]. On the 6th of January 2019, the CAG detected a 6.6-magnitude earthquake that occurred in Indonesia at 17:27 UTC. From the measured variations in acceleration over time, the vertical component of the Rayleigh wave was determined to have a period of ∼30 s and a peak-to-peak amplitude of ∼90 μm. These types of measurements are useful in the study and detection of earthquakes and suggest this could be a future use for CAGs.

Initial demonstrations of volcano monitoring were carried out on Mount Etna using the portable CAG from Exail Quantum Sensors, Paris, France (formerly iXblue) in 2022 [79,96]. The CAG was installed at an elevation of 800 m at the Pizzi Deneri Volcanological Observatory, approximately 2.5 km from the summit craters. During deployment, the CAG provided continuous gravity data allowing the tracking of volcano-related gravity changes with amplitudes ranging from tens to hundreds of nm s${}^{-2}$ over a wide range of time scales.

Environmental monitoring applications such as these have the potential to greatly improve resilience to geohazards through more accurate prediction and management of the risks. Furthermore, as performance improves, so does the potential to monitor other environmental signals such as water aquifers, allowing us to understand our changing environment and manage resources more effectively.

### 3.3. Small Scale Mapping for Engineering Applications

In 2022, a cold atom gravity gradiometer from the University of Birmingham was used to measure a utility tunnel under a road [97,98] with a signal-to-noise ratio of 8, locating the centre of the tunnel to within 20 cm. Such a result shows the technology is capable of having a transformative effect on reducing the risk of unforeseen ground conditions in the construction industry, as well as offering new mapping capabilities for archaeology, agriculture, natural resources, and defence capabilities.

### 3.4. Regional and Geological Scale Surveys

To perform larger scale regional and geological scale surveys, like those typically used in applications such as hydrology and oil prospecting, CAGs have been integrated into cars, trucks, ships, and planes.

The gravimeter GIRAFE from the Office National d’Etudes et de Recherches Aérospatiales (ONERA) has demonstrated dynamic measurements on ships, and planes. In 2018, the gravimeter achieved a shipboard measurement with a precision of 0.2–0.6 mGal under a 4 sea state condition and was used to map an area of the Meriadzec terrace located in the North Atlantic ocean [99]. In 2020, the GIRAFE system was used in an airborne campaign across Iceland, yielding gravity measurements with an estimated error of 1.7–3.9 mGal [100].

Truck-borne gravity mapping has been carried out using two different CAGs: the CAG from the University of California, Berkeley, and the CAG from Zhejiang University of Technology in Hangzhou. The former performed a gravity survey in the Berkeley Hills in 2019 achieving an uncertainty of 0.04 mGal, which allowed for the determination of subsurface rock density from the vertical gravity gradient [93]. In 2022 the CAG from Zhejiang University of Technology conducted a survey at the Xianlin Reservoir in Hangzhou. With internal and external coincidence accuracy of 35.4 μGal and 76.7 μGal, respectively, the results were verified by comparing the theoretical values obtained through forward modeling of a local high-resolution digital elevation model to the measured values, which showed good agreement [101].

In 2021, a car-mounted CAG from the Huazhong University of Science and Technology achieved the sensitivity of 1.9 mGal/$\sqrt{Hz}$ and the accuracy of more than 30 μGal in field measurements [102] while performing a survey on Yujia mountain.

In addition to these demonstrations, other CAGS are either under development [103] or have been developed, which could be used for large area mapping [104,105]. For example, the CAG from Huazhong University of Science and Technology in 2021 demonstrated operation in a moving vehicle. During this demonstration, it achieved a sensitivity of 60.88 mGal$\sqrt{Hz}$ with T = 5 ms [104] and could be used in the future for large-area gravity mapping.

It is clear that portable CAGs are progressing to devices capable of precise measurements while in motion and will enable their use in applications such as hydrology, oil and mineral prospecting, as well as navigation.

### 3.5. Space Based Systems

CAGs for use in space are currently under development [106,107,108,109,110] for several applications such as global gravity field mapping and fundamental physics [26]. For example, in fundamental physics CAGs are being developed with the aim to test the Weak Equivalence Principle [111,112]. While there is a lot of work in developing CAGS for use in space, there have been some existing cold atom systems demonstrated in space [113]. The first interference experiments were performed in 2017 during a space flight on the MAIUS-1 rocket [114]. Later in May 2018, NASA’s Cold Atom Laboratory (CAL) was launched to the International Space Station and has been operating onboard since then [115]. CAL is a quantum facility for studying ultra-cold gases in a microgravity environment and is being used to perform research in a force-free environment inaccessible to terrestrial laboratories, allowing for greater T times to be realised than practical in ground-based experiments.

## 4. Towards Improved Sensors and a Mature Commercial Market

### 4.1. The Commercial Market for CAGs

The current generation of portable CAGs is already sensitive and robust enough to be used in some applications with several systems already commercially available. Most notably, the systems from Exail (formerly iXblue) [94,116] made a massive contribution towards defining and accelerating the global quantum sensing market. Other commercial sensors are under development at companies including M Squared lasers [117], AOSense, Inc. [118,119], Mugaltech [120] and CASColdAtom [121]. Valued at $474.06 million in 2022, the market is expected to grow to over $740 million by 2028, with gravity applications being one of the largest market segments next to timing [122]. Whilst the market advances in a stable way with a predicted compound annual growth rate of 7.8%, there is a significant potential for further technological development and with it, significant market growth.

For example, making this technology robust enough to survive in harsh environments will enable quantum gravimeters to access the borehole market for geothermal applications. This is a fast-growing market valued in 2021 at $5.3 billion and is expected to grow to over $7 billion by 2030 [123]. Enabling long time continuous use of these quantum gravimeters together with potential large-area scanning capabilities are another advancement that can enable additional markets such as carbon capture and storage (>$4 billion [124]) and monitoring of geohazards (>$680 million [125]). Ultimately, increasing the performance, robustness, and ease of use of these systems could have a major impact on the hydrocarbon market. This is one of the largest markets this technology can address, valued at over $71 trillion in 2022 [126].

Overall, the opportunities enabled by commercial advances of CAGs are significant and by tailoring subcomponents of the system to address the size, weight, power, cost, user accessibility, and performance the technology could see a significant acceleration in market adoption.

### 4.2. Technological Routes to Improved Sensors

To facilitate the current generation of portable CAGs, several simplifications compared to lab systems and innovations have been required to meet the required robustness and SWaP. For example, to simplify the laser system, some CAGs have simply dropped atoms after the laser cooling and trapping stage, rather than utilise a fountain launch, which is commonly used in lab systems. This simplifies the laser system by requiring a single frequency of cooling light rather than the two or more frequencies needed for a fountain launch. To simplify the laser and vacuum systems, novel atom cooling and trapping geometries have enabled a reduction in system SWAP through a reduction in optical complexity compared with traditional 6-beam magneto-optical traps (MOTs). For example, pyramid MOTs [94,127], prism MOTs [97], mirror MOTs [128], and grating MOTs [129,130,131] all enable magneto-optical trapping with fewer input beams than traditional six beam MOTs. Several systems have also reduced the vacuum system and optical system size by using the same optical path between the MOT beams and interferometry beams [94,97,127].

It is expected that improvements in sensitivity, measurement speed, robustness, usability, autonomy as well as reduced size, weight, and power requirements will be achieved in the next generation of portable GAGs through the use of the latest and future techniques [132,133,134], technological developments [37,135,136,137,138,139,140,141,142], improved sub-components [143], systems engineering [144,145], and optimisation of sensor design [146,147]. Some examples of technology developments, along with the expected benefits, are highlighted in Table 2.

### 4.3. Physics Routes to Improved Sensors

While technological developments are key to bringing CAGs to market, advancing in the understanding and ability to manipulate cold atoms is also key to reaching the full potential of CAGs. In particular, realising techniques to increase the sensitivity of CAGs will be important to realising as small a form factor as possible.

The minimum length of a quantum gravity sensor is limited by the T time of the instrument, requiring a drop distance to allow the atomic wavepackets to evolve in time. As instruments are designed for smaller applications, to maintain or further improve the instrument sensitivity, one key technique that will need to be demonstrated in portable systems is large momentum transfer (LMT), which by applying sequential light pulses augments the Mach–Zender type interferometer by transferring multiple photon momenta to the cloud, increasing the shot noise limit sensitivity proportionally to the photon momenta [148,149,150,151]. In principle, devices can operate at reduced T time to achieve the same sensitivity, which would result in more compact instruments. Demonstrations in laboratory systems have shown in excess of 400ℏk of photon momentum in an interferometry sequence, using optimised pulse schemes to achieve high fidelity atom optics [152], which if implemented in portable gravimeters brings the sensitivity of atom interferometers far higher than their classical counterparts. Large momentum transfer is a key requirement for future atom interferometry dark matter and gravitational wave detectors, where photon momenta of >$10^{3}$ are required [153,154].

Techniques developed in achieving high fidelity atom optics from the fundamental physics applications should be applicable to future portable devices since the techniques are generally around light pulse engineering of the frequency, phase, and amplitude [155,156,157,158]. These schemes can introduce robustness to parameters such as detuning or intensity, reducing the effect of laser system noise on the atom interferometer. Some of these pulse schemes are applicable in LMT, and so may be implemented together in portable systems.

For high data rate applications, such as operation on moving platforms, single shot atom interferometry measurements will be beneficial to increase the spatial resolution of CAGs. Techniques such as phase shear have been used to read out the entire interferometer fringe in a single image by applying a tilt to the final interferometer pulse, creating a spatially varying interference pattern [133].

To reduce the power requirements of the laser system, cavity-enhanced atom interferometry may be useful in future portable CAGs, where atom interferometry occurs within an optical cavity [159]. Cavities are more stable and technologically simpler when smaller, which may lend themselves to compact sensor development. For larger and larger cavity systems, the cavity begins to hinder the atom interferometry pulses [159,160], requiring complicated schemes to overcome the cavity limits [161,162].

It is expected that several of the physics improvements described here will not be present in the first generation of portable systems, but in later generations of portable CAGs.

**Table 2 sensors-23-07651-t002:** Examples of technological developments which could be used to improve portable CAGs and their expected benefits.

Technological Development	Size	Weight	Power	Noise and/or Bias Reduction	Robustness	Refs.
Additive manufacturing (e.g., 3D printing)	✓	✓				[135,136,137,163,164]
Beam shaping (e.g, Top hat)				✓	✓	[139]
Clean atom sources				✓		[165,166,167]
Compact 2D MOTs	✓	✓	✓			[168,169]
Compact laser systems	✓	✓	✓			[170,171]
Micro-fabricated components	✓	✓	✓			[172]
Metasurfaces	✓	✓				[173,174]
Compact spectroscopy cells	✓	✓	✓			[175,176,177,178]
Optimised Coil systems	✓	✓	✓	✓	✓	[140,179,180]
Optimised electronics	✓	✓	✓	✓	✓	[181,182]
Passive vacuum systems	✓	✓	✓			[132,183]
Vacuum compatible anti reflection coatings				✓		[138]
Vibration Compensation				✓	✓	[104,142,184]

## 5. Conclusions

Portable atom interferometry-based gravity and gravity gradient sensors are becoming more prevalent, with some commercial devices entering the market [94,116,121] and many university systems under development [25,185]. While the current generation of portable CAGs is sensitive and robust enough for use in some applications, there is significant potential for further development. It is expected that improvements in sensitivity, robustness, and usability, as well as having reduced size, weight, and power requirements, will be achieved in future systems. These improvements will not only enhance the performance of the sensors but also enable their operation in new applications and on new platforms, such as unstaffed aerial vehicles [103,138], trains [25], cube satellites [186], and down boreholes [187].

## Figures and Tables

**Figure 1 sensors-23-07651-f001:**
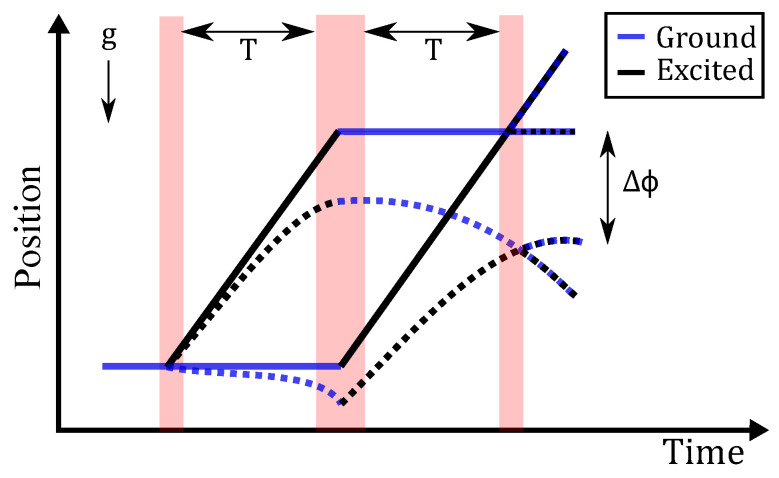
The atom interferometry sequence: As the atoms fall under the effect of gravity, a laser pulse is used to put them into a quantum superposition of the ground and excited states. The action of this pulse can be thought of as sending each atom along two simultaneous but different paths in the gravity field (as a consequence of wave-particle duality). After a time, T, a second laser pulse is then used to cause the paths to converge again at time 2T. At this time, the atoms are recombined using a final laser pulse; the final state of the atoms encodes the value of gravity. Dotted and solid lines show the sequence with and without gravity, respectively, where the deflection due to gravity introduces a phase shift Δϕ to the interferometer.

**Table 1 sensors-23-07651-t001:** Examples of the signal sizes across different applications.

Target	Signal Size (μGal)	References
Archaeology	15–40	[5,58,59]
Carbon storage monitoring	0–16	[60,61,62]
Cave detection and mapping	0–1500	[63,64,65]
Earth tide measurements	100–300	[66]
Earthquake detection	0–16,000	[6,67]
Hydrology	0–100	[68,69,70]
Mine shafts	0–100	[71,72]
Sinkholes	0–40	[73,74,75]
Tunnels	0–300	[19,76,77]
Volcano monitoring	0–60,000	[78,79,80]

## Data Availability

Not applicable.

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
