# Peer review of "Advances in Portable Atom Interferometry-Based Gravity Sensing"

_sensors, 2023, doi:10.3390/s23177651_

Round 1

Reviewer 1 Report

-Intro needs to be improved, with deeper state of the art review and explicit contributions to the study

-Operating principle; are there any supporting physics eqn's to go with it?

-The submission appears to be effectively a state of the art review? The writing needs to be revised to reflect this, alongside the concluding remarks

n/a

Author Response

Intro needs to be improved, with deeper state of the art review and explicit contributions to the study

To address this we have made the following changes to the manuscript:

  • We have added in details of the best sensitivity and resolution achieved with an atom interferometer to date into the introduction.
  • We have changed the final paragraph of the introduction to provide a more detailed and explicit description of the paper contents.
  • We have referenced additional papers to highlight the work that has gone on internationally in the field of atom interferometry and the progress towards the target applications listed.
  • We have added in details on the conversion from Gals to SI units to aid the reader.

Operating principle; are there any supporting physics eqn's to go with it?

We have included the equation for the phase shift and shot noise limited sensitivity as this will help the reader fully appreciate the discussion in section 4.3.

The submission appears to be effectively a state of the art review? The writing needs to be revised to reflect this, alongside the concluding remarks

We have changed the article type to review and modified the abstract, introduction and conclusion to reflect this

Author Response

This paper provides an overview of the current progress in the area of portable atom gravimeter and its applications. This is a hot topic. The contents are quite full and would be very interesting to readers in this area. There is some information for the authors, probably helpful to improve this review paper.

We are happy the reviewer finds the contents of the article to be of interest.

1. In ref. 98, the page number was missing, the complete info is “Sensors, 2022, 22(16): 6172.”

The missing information has been added to complete the reference.

2. On page 4, line 108, “the detection of earthquake: On the 6th of January 2019, the CAG detected a 6.6-magnitude earthquake 108 which occurred in Indonesia at 17:27 UTC”. Before this paper, a whole seismic wave of about 1 h that occurred in Pakistan on 28 September 2013 was recorded by the atomic gravimeter. [Bin Wu et al. -level cold atom gravimeter for field applications, 2014, Metrologia, 51 452.]

This information has been added to section 3.1 of the manuscript as suggested

3. From line 176-186, the commercial market for CAGs was discussed. For your information, a also producing atom gravimeter, the website is: www.mugaltech.com. If it is included, the information would be more complete.

Thank you for highlighting this.  We have added this information on the additional company to the manuscript.

Reviewer 3 Report

This manuscript is a commentary, meant to surmise the state of the art and to make forecasts on improvements and new applications. As such, this reviewer finds the manuscript a pleasure to read, considering that atom interferometry is among the most challenging physics experimentation. The writers are very knowledgeable in the field and convey to the reader the excitement of new challenges. The readers will benefit from the large survey of the article. The recommendation is that the journal should publish the commentary immediately.

Author Response

This manuscript is a commentary, meant to surmise the state of the art and to make forecasts on improvements and new applications. As such, this reviewer finds the manuscript a pleasure to read, considering that atom interferometry is among the most challenging physics experimentation. The writers are very knowledgeable in the field and convey to the reader the excitement of new challenges. The readers will benefit from the large survey of the article. The recommendation is that the journal should publish the commentary immediately.

We would like to thank the reviewer for their comments and we are glad they enjoyed the article.

Reviewer 4 Report

Good commentary article. Accept with minor modifications.

Comments:

1) See all comments and edits as notes in attached file (lines 56-60, 79, 97, 99, 173, 203, 209, and table 22 'optimized electronics, row)

2) Of above, required modifications:

a) Move first paragraph of section 3 (lines 56-60) to second paragraph before last in either section 1  or secttion 2.

b) Throughout  section 3.1. Metrology correct meteorology/meteorogical to metrology/metrological (lines 79, 97, 99).

c) Section 3.5. Space based systems line 173. The space environment does not provide temperature regimes unavailable on the ground.

d) Table 2, Optimized electronics line does not have the appropriate check marks.

Author Response

1) See all comments and edits as notes in attached file (lines 56-60, 79, 97, 99, 173, 203, 209, and table 22 'optimized electronics, row)

The attached comments have now been reviewed and have been corrected in line with the reviewers suggestions.

2) Of above, required modifications:

a) Move first paragraph of section 3 (lines 56-60) to second paragraph before last in either section 1  or secttion 2.

We have moved this paragraph into section 1 as requested.

b) Throughout section 3.1. Metrology correct meteorology/meteorogical to metrology/metrological (lines 79, 97, 99).

The mispellings have been corrected in line with the reviewers suggestion.

c) Section 3.5. Space based systems line 173. The space environment does not provide temperature regimes unavailable on the ground.

We have removed this statement and made a comment about how this allows greater T times to be achieved than is practical in ground based systems.

d) Table 2, Optimized electronics line does not have the appropriate check marks.

We have added check marks in the appropriate places.

Round 2

Reviewer 1 Report

Thanks for making the revisions